# Towards exploring current challenges and future opportunities relating to the prehospital triage of patients with traumatic brain injury: a mixed-methods study protocol

Naif Alqurashi,[1,2] Ahmed Alotaibi [iD] ,[2,3] Steve Bell,[4] Fiona Lecky [iD] ,[5] Richard Body[2]

For numbered affiliations see end of article.

**Correspondence to**
Naif Alqurashi;
nalqurashi1@ksu.edu.sa

## ABSTRACT

**Introduction** Traumatic brain injury (TBI) is a major global health burden that results in disability and loss of health. Identifying those patients who require specialist neuroscience care can be challenging due to the low accuracy of existing prehospital trauma triage tools. Despite the widespread use of decision aids to 'rule out' TBI in hospitals, they are not widely used in the prehospital environment. We aim to provide a snapshot of current prehospital practices in the UK, and to explore facilitators and challenges that may be encountered when adopting new tools for decision support.

**Methods and analysis** A mixed-methods study will be conducted using a convergent design approach. In the first phase, we will conduct a national survey of current practice in which every participating ambulance service in the UK will receive an online questionnaire, and only one response is required. In the second phase, semistructured interviews will be conducted to explore the perceptions of ambulance service personnel regarding the implementation of new triage methods that may enhance triage decisions. The survey questions and the interview topic guide were piloted and externally reviewed. Quantitative data will be summarised using descriptive statistics; qualitative data will be analysed thematically.

**Ethics and dissemination** This study has been approved by the Health Research Authority (REC reference 22/HRA/2035). Our findings may inform the design of future care pathways and research as well as identify challenges and opportunities for future development of prehospital triage tools for patients with suspected TBI. Our findings will be published in peer-reviewed journals, relevant national and international conferences, and will be included in a PhD thesis.

## STRENGTHS AND LIMITATIONS OF THIS STUDY

⇒ The proposed study incorporates a mixed-methods approach using a convergent design.
⇒ This study will employ a rigorous qualitative methodology in addition to a survey of current practice in order to map current practice within National Health Service ambulance trusts in the UK.
⇒ There will be some challenges that are related to the passion or enthusiasm of our participants about traumatic brain injury.
⇒ Recall bias may be a potential limitation.

centre (MTC), bypassing the nearest non-specialist acute hospital, which has been shown to positively impact outcomes.[2] Current guidelines from the National Institute for Health and Care Excellence guidelines recommend that all patients with severe TBI should be transferred to and treated in an MTC.[1 3] In a retrospective evaluation of over 22 000 patients with head injury, those who received initial care at a local hospital, rather than at an MTC with neurosurgical capability, had more than twice the odds of dying, after adjusting for casemix.[4]

Early neurosurgical intervention reduces mortality following TBI,[5] emphasising the importance of accurate identification of TBI in the prehospital environment. Field triage tools have been developed to provide rapid and accurate identification of patients with major injuries who require MTC care. Such tools typically use a combination of anatomic and physiological parameters together with injury mechanisms as predictive variables.[6] However, it can be challenging to identify patients with TBI in the prehospital setting as many patients initially present with relatively minor symptoms, but their condition

## BACKGROUND

Traumatic brain injury (TBI) remains one of the most common causes of death among young adults in the UK, and it is well recognised for causing long-term disability.[1] The establishment of major trauma networks in the UK enables prehospital care providers to transport patients with TBI in need of specialised trauma care to a major trauma

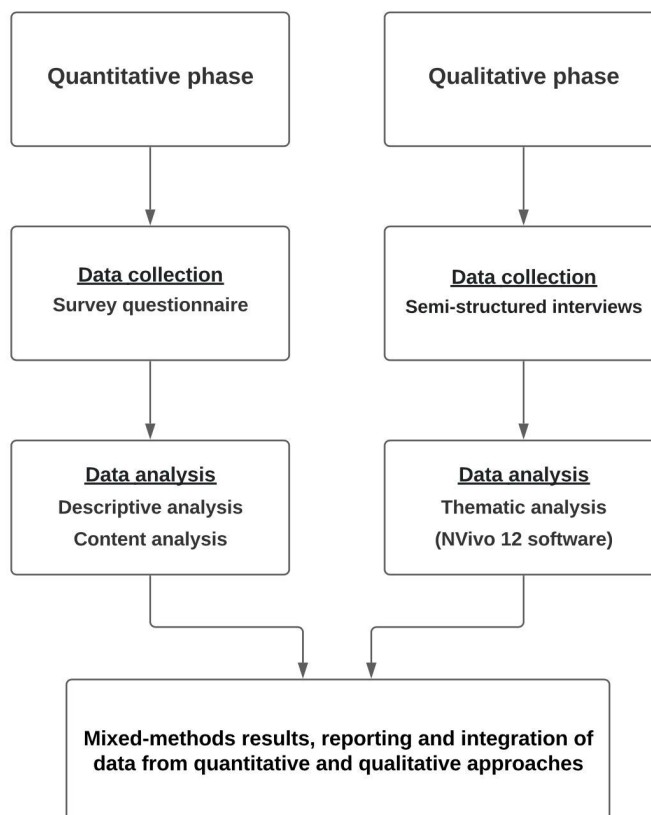

**Figure 1** An overview of the study design.

can deteriorate rapidly.[7] Triage decisions are also likely to be affected by confounders such as intoxication, temporary traumatic amnesia and age-related physiological responses to injuries.

Prior to planning this study, our group conducted a systematic review to evaluate the current prehospital triage tools for identifying patients with TBI. The sensitivity of previously validated triage tools ranged from 19.8% to 87.9% and specificity from 41.4% to 94.4%.[8] These values fall below the sensitivity target (>95%) and specificity target (>50%) of the American College of Surgeons Committee on Trauma,[9] emphasising the importance of optimising prehospital triage for patients with suspected TBI.

There is an increasing interest in the potential of using brain biomarkers in the emergency department and prehospital setting to identify and stratify patients with TBI. Additionally, prehospital triage could be optimised by measuring oxygen saturation in the brain using near-infrared spectroscopy (NIRS), which is a non-invasive technique that can be employed in the prehospital setting.[10] Recently, a number of studies have investigated the use of brain biomarkers, NIRS and clinical decision rules to enhance early diagnosis of TBI in the prehospital field.[11 12] Emergency physicians are currently using a number of clinical decision rules that have been developed to assess the necessity for cranial imaging in patients with mild TBI, who present with a GCS score of 13–15.[13 14] These include the New Orleans Criteria, the National Emergency X-Radiography Utilisation Study II and the Canadian CT head rule. However, none of these clinical

decision rules have been validated for use in prehospital triage.

Based on the evidence, it seems that there is a clinical need to gain in-depth insight into the current practice of prehospital TBI care in the UK and to identify potential gaps or areas for improvement needed to enhance prehospital triage. Therefore, a mixed-methods study will be conducted to provide a snapshot of the current practice and approaches to TBI care. It will also explore the main barriers and challenges, as perceived by prehospital clinicians, to adopting new decision aids that might enhance triage for patients with suspected TBI.

### Research aims

- ▶ To gain a deeper insight into current prehospital practice for triaging patients with suspected TBI in the UK.
- ▶ To determine whether prehospital care provides perceived gaps or areas for improvement in prehospital triage tools for patients with suspected TBI.
- ▶ To seek opinions from prehospital care providers about the feasibility of implementing clinical decision rules, brain biomarkers and NIRS in the prehospital setting.
- ▶ To identify the potential barriers to the implementation of new triage tools to enhance prehospital triage of patients with TBI.

## METHODS AND ANALYSIS
### Study design overview
A mixed-methods study using a convergent design will be conducted in which both quantitative and qualitative data will be collected, which will then be integrated during the interpretation phase.[15] Figure 1 shows the overall study process. In recent years, the mixed-methods approach has become increasingly popular in prehospital care research. In the literature, various variations of mixed-methods approaches have been described.[16 17]

This study is designed to be conducted in two phases. First, a web-based survey questionnaire will be conducted to collect quantitative data; then semistructured interviews will obtain qualitative data. The two phases will, however, be conducted concurrently and treated equally. The qualitative and quantitative results will be combined in the data interpretation phase to meet the aims of this study. The study protocol was developed by the research team after several formal discussions, and it was reviewed by an independent expert in mixed-methods studies. The data collection period started in June 2022, and it is expected to be completed by the end of March 2023. The research team is composed of individuals with sufficient experience and knowledge in emergency medicine and prehospital care.

### Theoretical framework
The philosophical foundation of mixed-methods studies remains a matter of considerable inquiry; however, it appears that pragmatism provides a useful framework for designing and conducting such studies.[18] This study will

be conducted in a pragmatic manner. Taking both quantitative and qualitative methods into account was essential for this study, and adopting a pragmatic epistemology provided the possibility of gaining new knowledge in an effort to improve prehospital triage for patients with suspected TBI. Additionally, the pragmatic approach does not adhere to a particular epistemology or reality, but rather selects methods in order to address the research question in the most appropriate way.

Considering the nature of the topic and the research objectives, this study will incorporate both constructivist and postpositivistic approaches. Ontological assumptions were made about the existence TBI as a condition (postpositivistic) and the current culture of paramedic practice (which, as a societal construct, is constructivist in nature). From an epistemological standpoint, we appreciate uncertainties and limitations that are associated with understanding the current approach. Embracing postpositivistic principles, we will seek to obtain a sample that is representative, though acknowledging the limitations of attempting broader generalisations to general clinical practice. Meanwhile, our use of qualitative methods to construct a framework by which we understand the nature of current clinical practice in this field is inherently constructivist in nature. By using this approach, complementary data can be collected, which will mitigate the weaknesses of each method of data collection.[19]

In the present study, it is preferable to adopt a concurrent triangulation design in which the quantitative and qualitative data collection procedures will be integrated in order to provide the opportunity to validate the findings of the two methods against each other. The decision to use this method was based on its potential to achieve a high level of rigour and enhance the significance of the findings.[20] Additionally, this approach broadens the scope of the study, providing a deeper insight into the topic.[21]

### Phase 1: quantitative data collection

In the first phase, a web-based survey questionnaire was developed to establish current prehospital triage practices for patients with TBI in the UK. In collaboration with the research team, an online survey questionnaire was developed and piloted among prehospital care personnel and paramedic PhD students to identify any potential issues with the design, the usability, content validity and duration required to complete the survey. The final version of the survey will be sent to each ambulance service that agreed to take part asking for one response per National Health Service (NHS) ambulance trust. Non-responding ambulance services will receive a reminder email after 3–6 weeks. The ambulance services will be contacted by telephone if they do not respond to the reminder email.

Participants will be required to fill out a short survey questionnaire consisting of three sections. The questionnaire was designed in accordance with good practice guidelines, taking into account specific elements such as layout, use of appropriate language, clarity of questions and the format of responses. The first section will include common demographic and employment questions, including the participant's level of experience, to identify the general characteristics of the participants. The second section will consist of targeted questions regarding whether any TBI-specific triage criteria are currently being used by the ambulance services in the UK; respondents will be asked to describe the current prehospital triage practices. In the third section, participants will be asked about their views and opinions regarding the implementation of point-of-care brain biomarkers, NIRS and the development of clinical decision rules in the future.

The questionnaire will include both closed-ended and open-ended questions, and Likert scales will be used to assess participant agreement with various statements. Participants will be asked to contact the research team if they are interested in participating in a semistructured interview. The quantitative data will be collected through a secure platform (Qualtrics-XM). It is expected that it will take approximately 12 min to complete the survey.

### Inclusion and exclusion criteria

The first phase of this study will involve a national survey of current practice. We will work with the research leads at each NHS ambulance trust in the UK via the National Ambulance Research Steering Group. Each research lead will be asked to identify a potential participant who would be willing to participate and represent the relevant NHS ambulance trust in this study. We require one survey response to reflect current practice within each NHS ambulance trust. The ambulance service will be asked to provide us with the name of one individual with the appropriate authority to complete the survey on behalf of the relevant NHS ambulance trust. Exclusion criteria will be limited to participants who will refuse to provide informed consent and do not complete the survey questionnaire.

### Survey data analysis

Each question will be analysed based on the number of participants who answer for each question. It is intended that the analysis will be descriptive and tabulated, and that quantitative survey data will be collected and analysed using Microsoft Office Excel (V.16.45). Open-ended questions and free-text responses will be analysed using simple content analysis to identify common themes, words and phrases.[22] A review of the survey responses will be conducted by NA and RB several times in order to determine whether there are any similarities or differences between the ambulance services.

### Phase 2: qualitative data collection

The second phase of this study will involve conducting semistructured interviews with prehospital care providers to obtain detailed information about their perspectives and beliefs about facilitators, potential barriers and the likely feasibility of implementing new decision aids to enhance prehospital triage for patients with suspected TBI. A semistructured

interview seems to be the best method for obtaining the data, as it would allow us to gain a more comprehensive understanding of a particular topic as well as capture the perspectives of our target population.[23] Through the interviews, it is hoped that we will be able to obtain rich data incorporating a range of descriptions that reflect different perspectives regarding the implementation of new decision aids for triaging patients with TBI.

The interview topic guide has been developed to meet the aims of the study (online supplemental appendices). The study team discussed, reviewed and approved the content of the interview topic guide. The interview topic guide was informed by the findings from our recent systematic review,[8] review of the literature and formal discussions with the study team. Participants will also be asked a series of open questions about their views of the need to improve the early identification of patients with TBI in the prehospital field. There are also questions which explore participants' opinions about the feasibility of introducing new technologies, such as biomarkers and NIRS, to identify patients with TBI, who require specialist neurosurgical care. The semistructured interview will consist of several open-ended questions; and to gain additional information, the interviewees' responses will guide follow-up questions. To reach the depth required to meet the study objectives, a series of questions to gain in-depth details will be asked, such as, 'Could you please clarify?', 'What do you mean?' and 'Can you give an example?'.

To ensure that the interview questions are accurate, understandable and consistent, the interview topic guide was piloted with PhD candidates.[24] The interviews will be recorded, transcribed and analysed using a thematic approach. Written consent will be obtained from each participant prior to recording the interviews. On request, a copy of the interview transcript will be provided to the participants for review to ensure accuracy and validity. The semistructured interviews will be conducted via video conference according to the participants' availability. It is anticipated that each interview will last approximately 30–60 min to achieve the depth required. Each participant will be assigned a code (participant ID) during the study process for confidentiality and anonymity purposes. Qualitative data will be reported in accordance with the Quality Standards for Reporting Qualitative Research.[25]

### Sampling method
We plan to invite each paramedic completing the survey to take part. Sampling will be purposive. As we anticipate a large degree of heterogeneity in practice between NHS ambulance trusts, we will seek representation from every trust in the UK. Once this sample has been obtained, we will analyse the demographics of the sample. We will continue to sample if there has been insufficient representation in any subgroups based on age, gender and level of experience. If more participants are required, participants will be asked to put a colleague in touch with us. Non-attendance and participant withdrawal from semistructured interviews will be the only criteria

for exclusion. It is estimated that a total of 15–20 semi-structured interviews will be required in order to reach the point of saturation, after which no new information is likely to be discovered during the process of data collection.[26]

### Qualitative data analysis
The interview transcripts will be analysed using the six-step thematic analysis approach proposed by Braun and Clarke.[27] After each interview session, the audio recordings will be anonymously transcribed, checked against the audio to ensure accuracy and read several times, word-by-word, to gain a general understanding of the qualitative data and to become familiar with all aspects of the data. The initial codes will be created, examined and subsequently categorised into subthemes and main themes. After the main themes are reviewed, defined and named, a report detailing the findings will be prepared. A 15-point checklist for conducting a good thematic analysis will be used to review the thematic analysis process and to ensure the reliability and quality of our thematic analysis.[27]

The interviews will be recorded by one author (NA), and to ensure the data quality, two independent authors (RB and AA) will compare randomly selected recording and their associated transcript. Similarly, the thematic analysis of the qualitative data will be performed by the first author and will be checked for accuracy by two other authors (RB and AA). In case of disagreement between the authors regarding initial codes, subthemes and main themes, formal discussions will be carried out to reach a consensus between authors. The four criteria proposed by Lincoln and Guba[28] will be employed to ensure that the data is of high quality and trustworthy, namely credibility, dependability, confirmability and transferability.[29] Different strategies will be employed to achieve credibility, including prolonged engagement with interviewees, member checking as well as ensuring a diverse sample so that multiple perspectives can be captured. It is our intention to spend sufficient time conducting each interview as well as processing each interview. Interviews will be conducted until saturation of data is achieved. To meet dependability, the written transcripts will be compared with the audio recordings, and a detailed description of the research methods will be provided. To ensure confirmability, an independent expert in qualitative research will review the analyses. In addition, a consensus will be reached among the members of the study team regarding the findings. To enhance transferability, the interview topic guide, the characteristics of the participants, as well as a detailed description of the selection process will be provided in order to allow other researchers to determine whether the findings are applicable to their own settings. Additionally, in line with the standard for enhancing rigour and credibility in qualitative research, a reflexive approach will be taken by the investigators. Transcripts will be interpreted by two separate investigators and coding will be undertaken by two investigators with different backgrounds. During this

process, investigators will dedicate time to discussing the potential impact of their own biases on the interpretation of the data.[30] The entire research team will appraise the coding framework, which offers a further opportunity to question any potential interference from inherent biases, and to examine issues of positionality. The study team includes members from relatively diverse backgrounds: a practising, research active paramedic; two academic (non-practising) paramedics; two emergency physicians; and one social sciences researcher. A sample of quotes that best represent each theme will be selected from the interview transcripts to ensure the credibility of the study findings. The qualitative data analysis will be conducted using the NVivo (V.12) software program.

### Ethical considerations

Approval to conduct this study has been granted by the Health Research Authority. Further, all ambulance services participating in the study have provided formal approval. As part of the recruitment process, participants will receive a written consent form and participant information sheet (PIS) detailing the study objectives, the data collection process and how confidentiality will be maintained. On the first page of the survey, participants will be asked to provide informed consent for taking part in the study. The survey will not progress beyond the first page if consent is not given. For the interview, the PIS and consent form will be sent to participating clinicians by email. They will be asked to complete the consent form and return it by email. Participants will have an opportunity to ask questions before deciding to take part in this study. The collected data will be stored in a secure location (a secure network) with access restricted only to the research team. Study group members will be responsible for monitoring the study progress, the data collection process and the credibility and integrity of the data.

### Patient and public involvement

During this phase of the research, we will examine current prehospital care practices and explore the opinions of ambulance clinicians regarding the feasibility of implementing different technologies. The results of this study will be used to inform the design of future research that will further assess the accuracy and safety of using different diagnostic technologies to diagnose TBI in the prehospital setting. It is our intention to work with patient and public representatives to design, manage, analyse and disseminate that research. We will also seek to involve patient and public representatives in the analysis and dissemination of this phase of the research.

### DISCUSSION

This study would be the first to provide a nationwide snapshot of the UK current prehospital triage practices for patients with suspected TBI. An important component of an effective trauma system is ensuring that appropriate patients are transported to an appropriate hospital.

Recently, research has identified the need for further studies that evaluate the benefits of using point-of-care brain biomarkers and NIRS for the rapid and accurate triage of patients with suspected TBI.[7 12] The mixed-methods approach has been widely used for advancing prehospital care in recent years and its use within the prehospital research field can help to explore research questions by collecting both quantitative and qualitative data.[20] In this study, the benefit of this approach is that it provides the opportunity to obtain a variety of opinions, perceptions and responses from prehospital clinicians, which in turn can enhance the depth of the quality of data as well as provide a national picture of the UK current prehospital care for triaging patients with suspected TBI.

The value of conducting semistructured interviews lies within the concept of eliciting richer in-depth data regarding the potential of applying new triage methods and exploring potential barriers and facilitators for improving prehospital triage of patients with suspected TBI. The present study will provide, for the first time, the perspectives of prehospital care providers towards implementing new triage tools to enhance prehospital triage pathways for patients with suspected TBI. Our national survey of current practice will establish the extent of variation in care pathways between UK ambulance services and gain in-depth insight into the decision aids that are currently being used to guide prehospital triage decision making in the UK.

It is intended that this study will inform the design of future research to evaluate further the accuracy and safety of using different triage tools to diagnose TBI in the prehospital environment. The study will directly inform the design of a feasibility study to evaluate the implementation of tools to improve prehospital triage of TBI. Our findings will help to determine the nature of the tool to be investigated (eg, clinical decision rule, biomarkers, NIRS) and how it may be applied by paramedics in real-world clinical practice.

A potential limitation in this study is that one interviewer might introduce bias into the coding of qualitative data; this is mitigated in the study design in having two other authors to review the coding process independently will help to ensure the accuracy of the data analysis. Considering that our survey was designed to understand the extent of variation in current prehospital practice for triage of patients with suspected TBI, there may be a potential for recall bias. Further, it is possible that participants may be more enthusiastic or passionate about TBI than non-participants. We will mitigate for this by inviting participants via the research lead for each NHS ambulance trust and by compensating paramedics for their time, regardless of their level of interest in TBI.

### Dissemination plan

On completion of the study, the results will be published in peer-reviewed medical journals, presented at relevant national and international conferences and will form part of a doctoral thesis (NA). The publication link will be

sent out to all study participates from the UK ambulance services. Our dissemination strategy also includes sharing the study findings with policy makers and experts in the prehospital field as well as via social media. The confidentiality of participants will be maintained throughout the dissemination process. It is hoped that the findings from this study will inform the design of future care pathways and provide a foundation for other research in this area.

**Author affiliations**
[1]Department of Accidents and Trauma, Prince Sultan bin Abdelaziz College for Emergency Medical Services, King Saud University, Riyadh, Saudi Arabia
[2]Division of Cardiovascular Sciences, University of Manchester, Manchester, UK
[3]College of Applied Medical Sciences, King Saud bin Abdulaziz University for Health Sciences, Riyadh, Saudi Arabia
[4]Medical Directorate, North West Ambulance Service NHS Trust, Bolton, UK
[5]School of Health and Related Research, University of Sheffield, Sheffield, UK

**Acknowledgements** We would like to express our appreciation for the guidance and support we received from the National Ambulance Research Steering Group during our initial efforts to contact Ambulance NHS Trusts in the UK.

**Contributors** NA prepared the original study protocol in collaboration with the study team (AA, SB, FL and RB). NA drafted the manuscript, and all coauthors (AA, SB, FL and RB) reviewed and approved it.

**Funding** The publication fee is funded by the cardiovascular department, University of Manchester, UK. No grant or award number.

**Competing interests** RB has consulted for Abbott Point of Care and is undertaking contract clinical research with BrainBox, which manufactures assays for biomarkers of TBI.

**Patient and public involvement** Patients and/or the public were not involved in the design, or conduct, or reporting, or dissemination plans of this research.

**Patient consent for publication** Not applicable.

**Provenance and peer review** Not commissioned; externally peer reviewed.

**ORCID iDs**
Ahmed Alotaibi http://orcid.org/0000-0002-6465-2687
Fiona Lecky http://orcid.org/0000-0001-6806-0921

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
