## [Reviewer comments · BMJ Open]

ARTICLE DETAILS

TITLE (PROVISIONAL)	Towards exploring current challenges and future opportunities relating to the prehospital triage of patients with traumatic brain injury: A mixed-methods study protocol
AUTHORS	Alqurashi, Naif; Alotaibi, Ahmed; Bell, Steve; Lecky, Fiona; Body, Richard

VERSION 1 – REVIEW

REVIEWER	Smith, Brandon NIHR Global Health Research Group on Neurotrauma
REVIEW RETURNED	08-Oct-2022

GENERAL COMMENTS	With interest I read your protocol for a mixed-methods study exploring the pre-hospital triage of TBI patients - it is excellent to see qualitative methodologies, of which remain sparingly used in this field, alongside quantitative. Overall this is a strong protocol, and I note a number of indicators of quality in every subheading; methodological detail is greatly appreciated, especially as mixed-methods efforts entail a great amount of information to report. The information presented is clear and coherent, and as a reader I am able to recognise the processes and flow that the study will take. In light of my personal expertise, my comments will primarily address the use of a 'Big Q' qualitative approach - reflexive thematic analysis. - I note that on Pg 12 In 22, the use of a 'good thematic analysis' checklist constructed by Braun and Clarke is used. Wary that this is a protocol, and that you may plan on fully deliberating on your methods in the results paper, I particularly would like to see more information on points 12-15 of this checklist, namely: The paradigmatic stance adopted by the authors in the conduct of the work. It is stated that mixed-methods is used - is this from a pragmatic paradigm often associated with mixed-methods research? How does/did this impact on your resultant epistemology/ontology/axiology/methodology? - Braun and Clarke note in a number of papers the importance of recognising one's paradigmatic stance. This is also found in the SRQR under point S5 . Is the same paradigmatic stance applied in both quantitative and qualitative components, or does it shift, i.e. from a post-positivist orientation (often associated with quantitative methods) to a interpretivist/constructivist orientation often seen in qualitative approaches? A brief rationale will be useful - an exploration of how one's paradigms influence their study is important
--

	in distinguishing from 'small q' and 'Big Q' research as advocated by Braun and Clarke (See Braun and Clarke 2019 'Reflecting on reflexive thematic analysis' - DOI 10.1080/2159676X.2019.1628806) - Establishing trustworthiness is mentioned (pg 11 ln 43), and rigour on pg 6 ln 52, but this is not elaborated further, which may be of interest for mixed methods/qualitative audiences, as as you may be aware, reliability/validity etc have little place in qualitative research. I note a reference by Nowell et al (2017) (10.1177/1609406917733847) that will be of use to address some of these concepts, namely credibility, transferability, dependability, confirmability, and largely of greater importance in a reflexive thematic analysis, how you plan to incorporate reflexivity in your research conduct. Will it be through personal accounts and reflexive journals? The importance of this may align with your paradigmatic stance, i.e., if that of an interpretivist, the relationship between your subjects and you, as the 'researcher as instrument' will be worth noting as it will influence your analysis. - In a similar notion towards establishing reflexivity, in point 15 of Braun and Clarke's checklist for a 'good' thematic analysis, research positionality is alluded to. In the SRQR also used in this study, I note point S6 'research characteristics and reflexivity'. Whilst you may mention this in your results paper, I am of the impression it is of equal importance to address points S5-S15 in any protocol papers, as these are often referred to in larger (mixed-methods) manuscripts in the interest of saving word count for results/discussion. What characteristics of the research team may influence your study? (particularly the lead authors and any authors partaking in qualitative inductive analysis) e.g., quali/MM experience, background in relation to the scope/context of the research questions - On line 45-50 of page 11, the line " A sample of quotes that best represent each theme will be selected from the interview transcripts to ensure the credibility of the study findings. " is given. In the previous line, it is mentioned that the synthesized results of each stage will be confirmed by the study team - is a similar mechanism planned such that initial themes will be shared with participants, in order to provide further opportunity to add further insights as to the research team's interpretation of findings? A useful reference for strategies to include rigour, such as member checking, can be found in Nicholls (2017, DOI 10.12968/ijtr.2017.24.3.114)
--	--

REVIEWER	Dumas, Ryan The University of Texas Southwestern Medical Center Medical School
REVIEW RETURNED	24-Nov-2022

GENERAL COMMENTS	Thank you for the opportunity to review this study protocol. In this study entitled "Towards exploring current challenges and future opportunities relating to the prehospital triage of patients with traumatic brain injury: A mixed-methods study protocol," the authors wish to describe and evaluate pre-hospital practices surrounding the triage and management of traumatically brain injured in the United Kingdom. To do this, a mixed-methods study will be performed that incorporates a survey questionnaire and in person interviews. A few comments:
--

	Study design overview 1. The authors note that the study protocol was reviewed by an independent expert in the field. Is this person a healthcare provider, a prehospital care provider, or someone who is not involved in patient care? I believe this distinction is important as it can introduce a specific bias into what is included in the study protocol. 2. Beyond internal circulation, has this survey been validated in any fashion? Participants Inclusion and Exclusion Criteria 3. Page 8, line 46-48, how do the investigators justify only completing one survey per company. It is unlikely that one person is able to accurately represent the provider-level variability and comfort managing TBI. What is the respondent is passionate about TBI and prehospital care, what is the respondent just started working as a paramedic? 4. How are the investigators account for experience? A new paramedic may be much more likely to be dependent on a tool for prehospital triage than a seasoned paramedic. 5. It seems as though only one person from each ambulance company will be designated to complete the online survey. The total number of participating centers is 12. With a small n, it will be hard to determine if there are any statistically significant conclusions to be made. Why not distribute to all crew members? 6. By only including one participant to complete the survey and one participant to undergo an interview, the companies may self-select for individuals who are more personally invested in the prehospital care of TBI patients and thus may bias the qualitative portion of this study. I doubt saturation will be reached through 12 interviews. Page 10, line 54, how do the investigators explain 15-20 interviews, yet interview will only be conducted to by those completing the survey (page 10, line 47). 7. How will the investigators ensure that the respondents are representative of national demographics of UK paramedics? Finally, there is no detailed description of how the statistics will be undertaken for this study. Please provide additional details regarding the plan for data analysis.
--	---

VERSION 1 – AUTHOR RESPONSE

Reviewer: 1:

C1: I note that on Pg 12 In 22, the use of a 'good thematic analysis' checklist constructed by Braun and Clarke is used. Wary that this is a protocol, and that you may plan on fully deliberating on your methods in the results paper, I particularly would like to see more information on points 12-15 of this checklist, namely: The paradigmatic stance adopted by the authors in the conduct of the work. It is stated that mixed-methods is used - is this from a pragmatic paradigm often associated with mixed-methods research? How does/did this impact on your resultant epistemology/ontology/axiology/methodology?

R1:

Thank you so much. In fact, we employed a convergent design (also known as a concurrent design), in which both qualitative and quantitative data are collected at the same time and then analysed separately but simultaneously. In this study, the aim is to map the current prehospital triage practices across the United Kingdom. In addition, the goal is to understand the extent of variation in current practice, whether any ambulance NHS trusts have a rule-out protocol for TBI (which may allow non-conveyance), and to gather preliminary thoughts about the likely feasibility of future innovations to enhance prehospital triage. The research philosophy in this study is based on a pragmatic approach that considers both positivists/ post-positivists and constructive paradigms. This approach involves selecting research methods that are relevant to the phenomenon under investigation rather than following a specific epistemology.

The study will be conducted in a pragmatic manner. Taking both quantitative and qualitative methods into account was essential for this study, and adopting a pragmatic epistemology provided the possibility of gaining new knowledge in an effort to improve prehospital triage for patients with suspected TBI.

The philosophical foundation of mixed-methods studies remains a matter of considerable inquiry; however, it appears that pragmatism provides a useful framework for designing and conducting such studies. As you know, the pragmatic approach does not adhere to a particular epistemology or reality, but rather selects methods in order to address the research question in the most appropriate way. The method section has been updated to include this information.

C2: Braun and Clarke note in a number of papers the importance of recognising one's paradigmatic stance. This is also found in the SRQR under point S5 . Is the same paradigmatic stance applied in both quantitative and qualitative components, or does it shift, i.e. from a post-positivist orientation (often associated with quantitative methods) to a interpretivist/constructivist orientation often seen in qualitative approaches? A brief rationale will be useful - an exploration of how one's paradigms influence their study is important in distinguishing from 'small q' and 'Big Q' research as advocated by Braun and Clarke (See Braun and Clarke 2019 'Reflecting on reflexive thematic analysis' - DOI 10.1080/2159676X.2019.1628806)

R2:

Thank you so much. In the present study, both methods are considered equally important, since the quantitative part will provide insight into prehospital care providers' views and needs. This was mentioned in the first version of the study protocol. Both quantitative and qualitative data are

complementary. The results obtained from both methods will then be synthesized to gain a deeper understanding of the issue being investigated.

C3: Establishing trustworthiness is mentioned (pg 11 ln 43), and rigour on pg 6 ln 52, but this is not elaborated further, which may be of interest for mixed methods/qualitative audiences, as as you may be aware, reliability/validity etc have little place in qualitative research. I note a reference by Nowell et al (2017) (10.1177/1609406917733847) that will be of use to address some of these concepts, namely credibility, transferability, dependability, confirmability, and largely of greater importance in a reflexive thematic analysis, how you plan to incorporate reflexivity in your research conduct. Will it be through personal accounts and reflexive journals? The importance of this may align with your paradigmatic stance, i.e., if that of an interpretivist, the relationship between your subjects and you, as the 'researcher as instrument' will be worth noting as it will influence your analysis.

R3:

Thank you so much. I added "The four criteria proposed by Lincoln and Guba will be employed to ensure that the data is of high quality and trustworthy, namely credibility, dependability, confirmability, and transferability". I have provided a detailed explanation of each one in the revised version.

C4: In a similar notion towards establishing reflexivity, in point 15 of Braun and Clarke's checklist for a 'good' thematic analysis, research positionality is alluded to. In the SRQR also used in this study, I note point S6 'research characteristics and reflexivity'. Whilst you may mention this in your results paper, I am of the impression it is of equal importance to address points S5-S15 in any protocol papers, as these are often referred to in larger (mixed-methods) manuscripts in the interest of saving word count for results/discussion. What characteristics of the research team may influence your study? (particularly the lead authors and any authors partaking in qualitative inductive analysis) e.g., quali/MM experience, background in relation to the scope/context of the research questions

R4:

Thank you so much. The study team consists of two professors of emergency medicine, two PhD paramedic students, and one consultant paramedic from North West Ambulance Services with excellent experience working together to plan this study protocol, design questionnaires, collect data, monitor study progress, and analyse quantitative and qualitative data. Members of the study team will engage in continuous discussion and reflection in order to ensure the credibility of the study findings. There will be a continual awareness of the impact our clinical and research backgrounds may have on the analytical process and on the discussion of the results. We have addressed that we will practice reflexivity in our response to the previous comment and we have added some discussion around

positionality, discussing the diverse backgrounds of our team and the importance of reflection in the analysis phase.

C5: On line 45-50 of page 11, the line " A sample of quotes that best represent each theme will be selected from the interview transcripts to ensure the credibility of the study findings. " is given. In the previous line, it is mentioned that the synthesized results of each stage will be confirmed by the study team - is a similar mechanism planned such that initial themes will be shared with participants, in order to provide further opportunity to add further insights as to the research team's interpretation of findings? A useful reference for strategies to include rigour, such as member checking, can be found in Nicholls (2017, DOI 10.12968/ijtr.2017.24.3.114)

R5:

Thank you so much. To make sure that the results of the study are accurate and reliable, we will utilise the member checking technique. We covered this in the revised version when we discussed the four criteria proposed by Lincoln and Guba for ensuring that the data is reliable and trustworthy (credibility, dependability, confirmability, and transferability).

Reviewer: 2:

C1: The authors note that the study protocol was reviewed by an independent expert in the field. Is this person a healthcare provider, a prehospital care provider, or someone who is not involved in patient care? I believe this distinction is important as it can introduce a specific bias into what is included in the study protocol.

R1:

Thank you so much. The study protocol was reviewed by an independent expert in qualitative research.

C2: Beyond internal circulation, has this survey been validated in any fashion?

R2:

Thank you so much. The survey has been validated to evaluate the survey content for possible outcomes such as clarity, accuracy, consistency, relevance, etc. Initially, as the lead author, I reviewed the components of the survey for content, flaws, and clarity. Next, the survey content was then reviewed by each member of our research team. I then sought peer review from PhD candidates in the same field in order to critically evaluate the relevance of the survey content. Some ambulance services requested the survey content as part of the approval process. The pilot phase was conducted using the Qualtrics platform, which will be used by the participants as well.

C3: Page 8, line 46-48, how do the investigators justify only completing one survey per company. It is unlikely that one person is able to accurately represent the provider-level variability and comfort managing TBI. What is the respondent is passionate about TBI and prehospital care, what is the respondent just started working as a paramedic?

R3:

The study team reflected on this question extensively during the design phase. Our goal is to ensure that the findings are transferrable and, to a large extent, representative of practice across the United Kingdom. While we acknowledge that no single participant could answer on behalf of every paramedic in a given ambulance service, having representation from every ambulance trust in the UK allows us to at least take account of much of the anticipated heterogeneity based on geography. Further, this method allows us to obtain the trust guidance from every NHS ambulance trust in the country. As we expect the guidance to be trust-wide, a single participant will be able to adequately represent the ambulance trust for this element.

It is possible that participants may be more enthusiastic about TBI than non-participants. This is always a risk in qualitative research. However, we mitigated for this by approaching the research leads at each ambulance NHS trust. We did not specifically ask for a participant who had interest in TBI. Further, we compensated participating paramedics for their time, regardless of whether they were interested in TBI. However, we still recognise that there is potential for some bias that cannot be mitigated, and we have added this to the limitations section.

C4: How are the investigators account for experience? A new paramedic may be much more likely to be dependent on a tool for prehospital triage than a seasoned paramedic.

R4:

Thank you so much. We are collecting data about the level of experience of participating paramedics, which means that we will be able to examine the data for any potential impact of experience. We have added this to the methods section.

C5: It seems as though only one person from each ambulance company will be designated to complete the online survey. The total number of participating centres is 12. With a small n, it will be hard to determine if there are any statistically significant conclusions to be made. Why not distribute to all crew members?

R5:

Thank you so much. The aim with this aspect is to map current practice on an ambulance NHS trust level. We don't require statistical power to make statistically significant conclusions. We expect that the data obtained from the surveys will lack some richness and we are concerned that it would be unfeasible to achieve a reasonable response rate with a wider survey.

C6: By only including one participant to complete the survey and one participant to undergo an interview, the companies may self-select for individuals who are more personally invested in the prehospital care of TBI patients and thus may bias the qualitative portion of this study. I doubt saturation will be reached through 12 interviews. Page 10, line 54, how do the investigators explain 15-20 interviews, yet interview will only be conducted to by those completing the survey (page 10, line 47).

R6:

Thank you so much. A saturation approach to data collection will be used in the qualitative phase of this mixed-methods study. We plan to invite each paramedic completing the survey to take part. If we happen to need any more, we may ask them to put a colleague in touch with us.

We agree with the reviewer that 12 interviews might be sufficient to reach data saturation, however, in this study we are not interviewing homogenous groups (different ambulance services (England, Wales, Scotland and Northern Ireland) and different roles (paramedics and critical care paramedics) as well as the heterogeneity of TBI (mild – moderate – severe).

C7: How will the investigators ensure that the respondents are representative of national demographics of UK paramedics?

R7:

Thank you so much. Our sampling strategy is essentially purposive, seeking to ensure diverse representation based on factors that may influence responses. The key factor that we wish to control for is the participant's employer (ambulance NHS trust) because we are aware that clinical practice is likely to be heterogeneous between ambulance NHS trusts. However, we also plan to take a purposive approach to ensuring adequate representation from participants based on age, gender, ethnic origin and years of experience. If, having completed our initial round of sampling, we have not achieved sufficiently diverse representation based on these factors, we will continue to sample until this has been addressed. This has now been added to the methods section.

C8: Finally, there is no detailed description of how the statistics will be undertaken for this study. Please provide additional details regarding the plan for data analysis.

R8:

Thank you so much. We will analyse survey data using descriptive methods. We do not plan to test any statistical hypotheses but rather present a summary of the data. We will use content analysis to analyse the data from open-ended survey questions and we will use thematic analysis to analyse interview data. We do not plan to undertake any statistical testing. Our approach to analysing survey data is described on page 9, whereas our approach for qualitative data is described on pages 11-13.

VERSION 2 – REVIEW

REVIEWER	Dumas, Ryan The University of Texas Southwestern Medical Center Medical School
REVIEW RETURNED	02-Feb-2023
GENERAL COMMENTS	The authors did a very nice job addressing this reviewer's concerns and the manuscript is now suitable for publication.